# Dietary folate intake and all-cause mortality and cardiovascular mortality in American adults with non-alcoholic fatty liver disease: Data from NHANES 2003 to 2018

**Jinsheng Dong**[1], **Zhiqiang Li**[2], **Chenlu Wang**[2], **Runshun Zhang**[1], **Yilin Li**[1], **Mingkun Liu**[1], **Qiuye Chen**[1], **Yuning Bai**[1]*, **Wenliang Lv**[1]*

1 Guang'anmen Hospital, China Academy of Chinese Medical Sciences, Beijing, China, 2 Beijing University of Chinese Medicine, Beijing, China

* bynedu@163.com (YB); lvwenliang@sohu.com (WL)

## Abstract

### Background

The relationship between dietary folate intake and prior mortality in adult patients with Non-alcoholic Fatty Liver Disease (NAFLD) has not been clearly studied. We aimed to examine the relationship between dietary folate intake and all-cause and cardiovascular (CVD) mortality in adult NAFLD patients in the US.

### Methods

Using data from National Health and Nutrition Examination Survey (NHANES) 2003–2018 and associated mortality data we conducted a cohort study of US adult NAFLD subjects. Multivariable Cox proportional hazards regression models were used to evaluate the relationship between dietary folate intake and both all-cause mortality and CVD mortality, accounting for potential confounders. The study employed restricted cubic spline analysis to investigate the non-linear association between dietary folate levels and mortality from all causes and cardiovascular disease.

### Results

Our final cohort consisted of 3,266 NAFLD patients, with a median follow-up of 10.3 years, 691 deaths were observed, including 221 cardiovascular deaths. Compared to participants with a folate intake in Quartile 1 ($\leq$250 µg/d), those in Quartile 4 ($\geq$467.5 µg/d) had multivariable-adjusted hazard ratios of 0.69 (95% CI, 0.51–0.94) for all-cause mortality (p for trend = 0.028) and 0.55 (95% CI, 0.29–1.04) for CVD mortality (p for trend = 0.107). A non-linear relationship between dietary intake and risk of death was not observed.

### Conclusion

Greater dietary folate intake is associated with a reduced risk of all-cause in American adults with NAFLD. Higher dietary folate intake not found to be associated with lower CVD

**Data Availability Statement:** All relevant data are within the Supporting Information files.

**Funding:** This study was supported by the Scientific and Technological Innovation Project of China Acad-emy of Chinese Medical Sciences(no. CI2021A00806). he funders had no role in study design, data collection and analysis, decision to publish, or preparation of the manuscript.

**Competing interests:** NO authors have competing interests.

mortality. These findings suggest that dietary folate may improve the prognosis of adult NAFLD patients. The measured-response relationship between dietary folate intake and mortality in patients with NAFLD requires further investigation.

## Introduction

Non-alcoholic Fatty Liver Disease (NAFLD) is a medical condition where there is an abnormal buildup of fat in liver cells, which is diagnosed by ruling out excessive alcohol consumption and other established causes of liver damage [1]. According to reports, the worldwide preva-lence of NAFL is considered to be approximately 30% [2], with an overall mortality rate of 15.44 per 1,000 person-years (ranging from 11.72 to 20.34) [3]. Over the past three decades, the total number of deaths among patients with non-alcoholic fatty liver has doubled [4]. Extrahepatic diseases such as cardiovascular disease and extrahepatic malignancies contribute significantly to the mortality of NAFLD patients [5]. Hence, it is crucial to investigate interven-tions and mechanisms aimed at decreasing the mortality rate among individuals with NAFLD.

Folate (vitamin B9), is a collective term for a group of compounds within the water-solu-ble vitamin family. This nutrient is essential for protein synthesis cell division and growth, which are crucial for the appropriate development of red blood cells [6]. A folate deficiency can lead to macrocytic anemia and increased amounts of homocysteine in the blood, which raises the risk of arteriosclerosis, thrombosis, and hypertension [7]. In an animal experi-ment, it was found that dietary intake of folate can improve the severity of nonalcoholic steatohepatitis (NASH) in NAFLD patients, thereby slowing down the progression of NAFLD [8]. Some retrospective studies have also shown a negative connection between the consumption of folate in one's diet and the folate levels in the blood serum in individuals with NAFLD [9–11].

Given the high prevalence and mortality of patients with NAFLD, it is important to find means to improve their prognosis. However, there hasn't yet been any prospective research done to evaluate the correlation between folate consumption and mortality in NAFLD patients. To fill this void in research, our study employed a sample of U.S. people that accu-rately represents the entire nation to examine the correlation between the intake of dietary folate in patients with NAFLD and the risk of all-cause mortality as well as cardiovascular dis-ease (CVD) mortality.

## Method

### Research design and subjects

The data used in our analysis was gathered from the National Health and Nutrition Examina-tion Survey (NHANES), a U.S. program assessing health and nutrition. NHANES utilizes a sophisticated, multi-stage probability design for the purpose of sampling. The data, publicly available and de-identified for privacy, was sourced from the NHANES database. During the data collection period or after, the authors did not have access to any information that could identify individual participants. Every participant gave their written consent, fully informed, at the time of enrollment [12]. The NHANES study protocols were approved by the Institutional Review Board of the National Center of Health Statistics. This research adhered to the Strengthening the Reporting of Observational Studies in Epidemiology (STROBE) standards for cohort studies to ensure thorough and clear reporting [13].

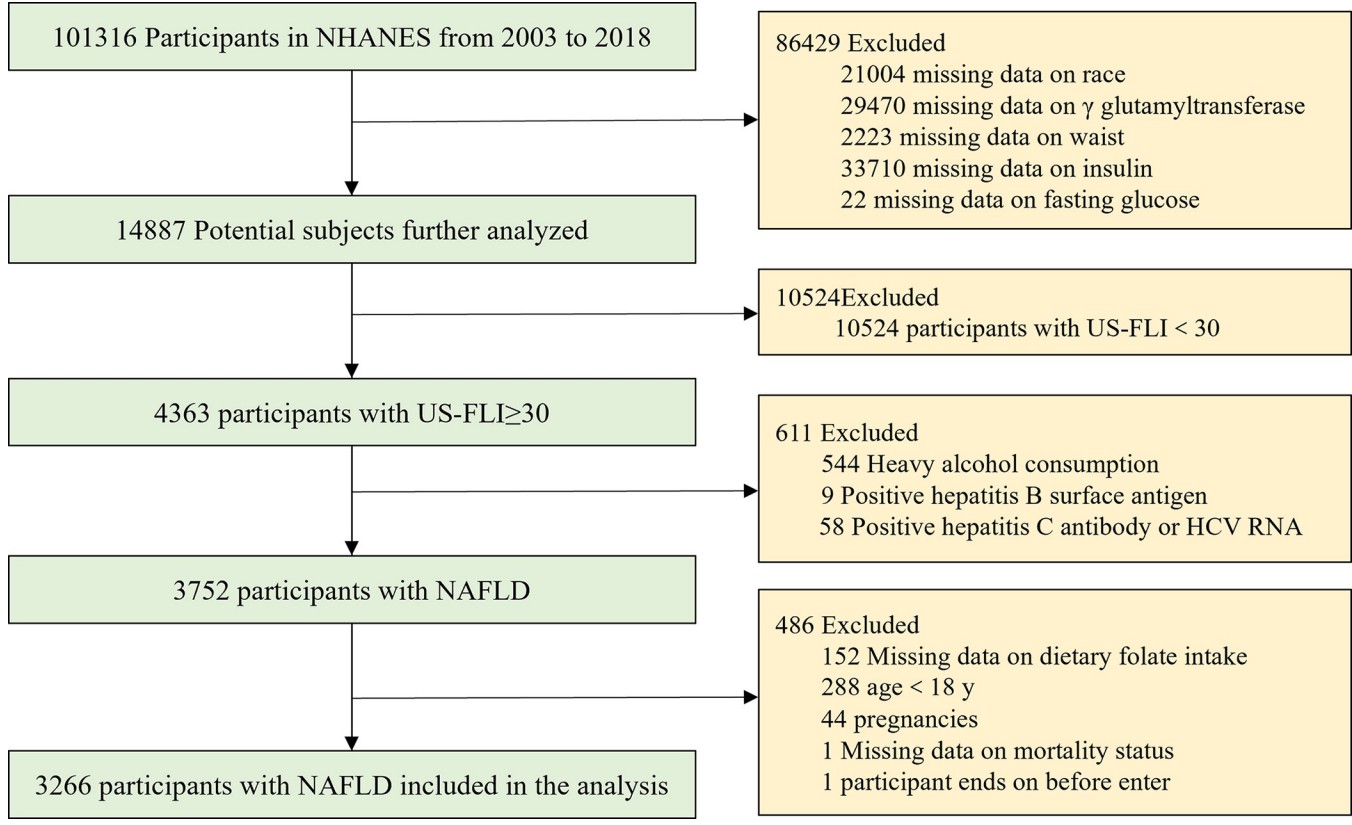

**Fig 1. Flowchart depicting the process of selecting the study population.**

The dataset was accessed on April 21, 2024, for research purposes. We initially identified 101,316 participants from eight cycles (2003–2018) of the NHANES study. After removing 86429 participants with incomplete data and 10524 participants with a US-FLI score below 30, we were left with 4,363 participants whose US-FLI ≥ 30. Further exclusions for heavy alcohol consumption(males alcohol intake greater than 30g/d and females alcohol intake greater than 20g/d), hepatitis B and C, resulted in a cohort of 3,752 participants NAFL. Finally, we excluded 486 participants due to missing data on dietary folate intake, age less than 18 years, pregnancy, missing data on mortality status, or who exited before the study began. As a result, the analysis encompassed a final cohort of 3,266 patients with NAFLD (Fig 1).

## Definition of NAFLD

We employed the US Fatty Liver Index (US-FLI) as a means of establishing and diagnosing NAFLD. The US-FLI is a non-invasive diagnostic tool that incorporates metabolic markers and demographic factors into a complex formula:

$US - FLI$

$$= \frac{e^{(-0.8073 \times \text{non-Hispanic Black} + 0.3458 \times \text{Mexican American} + 0.0093 \times \text{age} + 0.6151 \times \log_e \gamma \text{ glutamyltransferase} + 0.0249 \times \text{waist circumference} + 1.1792 \times \log_e \text{ insulin} + 0.8242 \times \log_e \text{glucose} - 14.7812)}}{1 + e^{(-0.8073 \times \text{non-Hispanic Black} + 0.3458 \times \text{Mexican American} + 0.0093 \times \text{age} + 0.6151 \times \log_e \gamma \text{ glutamyltransferase} + 0.0249 \times \text{waist circumference} + 1.1792 \times \log_e \text{ insulin} + 0.8242 \times \log_e \text{ glucose} - 14.7812)}}$$
$$\times 100$$

If a participant identified as non-Hispanic Black or Mexican American, their values were set to 1, and if not, they were set to 0. Measurements included age in years, international U/L for γ glutamyltransferase, cm for waist circumference, pmol/L for insulin, and mg/dL for glucose [14]. After eliminating those who engaged in excessive alcohol consumption (males alcohol intake > 30g/d and females alcohol intake>20g/d) and hepatitis B and C, Patients whose US-FLI score was 30 or higher were classified as having NAFLD. This method aligns with recent research suggesting that the US-FLI provides a reliable, cost-effective method for diagnosing NAFLD in large-scale epidemiological studies [15, 16].

## Dietary folate intake measurements

The assessment of dietary folate consumption was conducted using the What We Eat in America (WWEIA) component of the NHANES. The WWEIA is a collaboration between the U.S. Department of Agriculture (USDA) and the U.S. This component collects detailed dietary intake information, including the specific kinds and amounts of food and beverages that were ingested in the 24 hours before the interview. The WWEIA uses the Automated Multiple Pass Method (AMPM), which is a completely computerized recall method that employs a 5-step interview process to ensure efficient and accurate collection of intakes. The AMPM has undergone validation in many investigations and has demonstrated its efficacy in accurately measuring the group energy intake of individuals [17, 18]. Every NHANES participant underwent two 24-hour dietary recall interviews, one conducted face-to-face and the other via telephone, with a time interval of 3 to 10 days. For our study, we utilized the daily summaries of food energy and 64 nutrients/food components, which encompassed dietary folate intake. These summaries were derived using the USDA's Food and Nutrient Database for Dietary Studies [19]. The study determined the daily dietary folate consumption by computing the mean of two dietary recalls provided by the individuals. For those with only one recall (9.15% of the 3266 participants), the single recall value was used.

## Ascertainment of mortality

We ascertained mortality status by cross-referencing participant data with the National Death Index records. The cause of death was determined using the ICD-10 codes. CVD mortality refers to deaths caused by heart disorders (codes I00–I09, I11, I13, and I20–I51) and cerebrovascular diseases (codes I60–I69). All-cause mortality encompassed deaths resulting from any cause. The follow-up period extended from the participant's visit to the NHANES Mobile Examination Center (MEC) until either the participant's death or the conclusion of the follow-up period on December 31, 2018, whichever transpired first.

## Covariates determination

Covariates were extracted for each participant to control for potential confounding factors. Age was divided into groups. Participants were categorized by race as Mexican American, Non-Hispanic White, Non-Hispanic Black, Other Hispanic, or Other Race. Education level was classified as less than high school, high school, or college or above. The income level was calculated by dividing the family's income by the poverty threshold and classifying it as either less than 1, between 1 and 3, or greater than 3. Participants who had engaged in vigorous or moderate physical activity in the past 30 days were considered physically active; otherwise, they were considered physically inactive. According to WHO standards, body mass index (BMI) was categorized as less than 25, 25–29.9, 30–34.9, 35–39.9 or 40 and above. The smoking status was divided into three parts: never smokers, former smokers, and current smokers [16]. Dietary supplement use was determined by the question 'Have you used or taken any vitamins,

minerals, or other dietary supplements in the past month?'. History of hypertension, diabetes, and cancer was obtained from questionnaires asking whether the participant had been diagnosed with these diseases by a doctor. Dyslipidemia is characterized by the presence of any of the following conditions: total cholesterol concentration 200 mg/dL or higher, low-density lipoprotein cholesterol concentration 130 mg/dL or higher, triglyceride concentration 150 mg/dL or higher, or a concentration of high-density lipoprotein cholesterol < 40 mg/dL [20]. Cardiovascular disease history was determined by questionnaire responses regarding congestive heart failure, coronary heart disease, angina/angina pectoris, or heart attack. Biochemical measurements included low-density lipoprotein cholesterol (LDL-C), high-density lipoprotein cholesterol (HDL-C), total cholesterol (TC), and alanine aminotransferase (ALT) levels, which were obtained from blood samples drawn under standard MEC procedures. Castelli_II index is the ratio of LDL-c to HDL-c, which better reflects the risk of cardiovascular disease (less than 3 is considered a low risk of cardiovascular disease) [21]. Methotrexate and metformin use may affect folate metabolism, and their use comes from prescription medications questionnaires. Daily sugar intake is from the same source as folate intake and is categorized as low (≤50 g/day) and medium and high (>50 g/day) according to WHO recommendations [22].

## Statistical analysis

In accordance with the analytic guidelines of NHANES, we took into account sample weights, stratification, and clustering to extrapolate our findings to the entire US population aged 18 years and above. To obtain suitable sampling weights for the comprehensive analysis of eight survey cycles, we split the Full Sample 2 Year MEC Exam Weight by 8. The data are depicted as medians (with interquartile ranges) for continuous variables and as numerical values and percentages for categorical variables. We Divide the dietary folate intake into four groups according to the quartiles. To compare group differences, the Chi-Square (X2) test was used for categorical data, the One-Way ANOVA test was used for variables with a normal distribution, and the Kruskal-Wallis H test was used for variables with a skewed distribution. These tests were employed to identify variations across different dietary folate intakes, specifically categorized into quartiles. In order to investigate the association between dietary folate consumption and the likelihood of mortality, we constructed three weighted Cox proportional hazards models using univariate and multivariate Cox regression models, including model 1 (did not adjust for any covariates,), model 2(only Sociodemographic variables: sex, age, and race/ethnicity) and model 3 (fully-adjusted model: model 2 plus education level, income level, BMI, hypertension, cardiovascular disease, cancer, physical activity, use of dietary supplements, diabetes, smoking status, dyslipidemia, alt, TC, Castelli_II index, use of methotrexate, use of methotrexate, daily sugar intake). Only covariates with significant p-values in the univariate analysis can be included in the fully-adjusted model. The Wald test was employed to examine the regression coefficients of each variable, and no violations were found. We documented the effect sizes along with their 95% confidence intervals.

In order to consider the non-linear correlation between dietary folate intake and the two kinds of mortality, we employed restricted cubic spline analysis to account for nonlinearity. A likelihood ratio test was performed to evaluate the presence of nonlinearity. The analyses were further stratified by subgroups based on race, gender, age, education levels, BMI, income bracket, smoking habits, presence of diabetes, hypertension, history of cardiovascular disease, dyslipidemia, level of physical activity, and daily sugar intake. The significance of interactions was evaluated by calculating the p-value for interaction.

To ensure the reliability of our findings, we performed a sensitivity analysis using the following procedures:(1) dietary folate intake was utilized as the continuous variable data, we

computed the P value to confirm the outcomes and to explore any possible nonlinearity;(2) to reduce the likelihood of reverse causation bias, we omitted individuals who passed away within a two-year follow-up period;(3) the sensitivity analyses did not include participants who had only one dietary folate intake recall;(4) we performed an additional analysis, excluding NAFLD patients who passed away within two years of follow-up and had only one dietary folate intake recall. This helped us further validate our results.

We employed the Random Forest Multiple Imputation to fill in the missing data in the covariates. The analysis of the data was performed using stata18 and the online analysis platform, light-scholar (https://www.light-scholar.com/). The tests were conducted using a two-sided methodology, and a P-value below 0.05 was deemed statistically significant.

## Results

### Baseline characteristics

In our research, we integrate a total of 3,266 confirmed cases of NAFLD. The weighted average age of the participants was 55.33 years. The cohort comprised 1,805 males (55.3%) and 1,461 females (44.7%). We categorized the participants into four quartiles based on their dietary folate intake levels: Quartile 1 ($\leq$250μg/d), Quartile 2 (250.5–342.5μg/d), Quartile 3 (343–467μg/d), and Quartile 4 ($\geq$467.5μg/d). The attributes of the NAFLD participants, categorized according to these quartiles, are outlined in Table 1. Notably, participants with higher dietary folate consumption, as opposed to those in the lowest quartile, were characterized by younger age, higher likelihood of being male, greater physical activity, better levels of education and income, and the prevalence of diabetes was lower among these participants. Biochemical parameters also varied significantly across the quartiles. ALT levels notably increased with higher dietary folate intake. Other parameters, including aspartate aminotransferase (AST), HDL-C, and TC, also showed significant variations across the quartiles. The levels of BMI, smoking status, hypertension, cardiovascular disease, cancer, dyslipidemia, LDL-C, and triglycerides (TG) were similar across all four quartiles of folate intake.

### Dietary folate intake and all-cause and CVD mortality

Over a median follow-up period of 10.3 years, ranging from 8.3 to 12.8 years, a total of 691 deaths were observed. Out of these, cardiovascular disease was the cause of 221 cases. For all-cause mortality, In Model 1, the hazard ratio (HR) for all-cause mortality in the fourth quartile ($\geq$467.5μg/d) was 0.57 (95% CI, 0.44–0.75) compared to the first quartile, indicating a significant decreasing trend (P< .0001). After making adjustments for multiple variables in Model 3, the HR for all-cause mortality in the fourth quartile was 0.69 (95% CI, 0.51–0.94) compared to the first quartile, still showing a significant decreasing trend (P = .028). For CVD mortality, In Model 1, the HR for the fourth quartile was 0.39 (95% CI, 0.24–0.64) compared to the first quartile, showing a significant decreasing trend (P < .0001). After adjusting for multiple variables in Model 3, the HR for CVD mortality in the fourth quartile was 0.55, with a 95% CI of 0.29–1.04. The decrease in trend did not reach statistical significance (P = .107) (Table 2). Fig 2A demonstrates a decreasing trend in the likelihood of death from any cause with increasing dietary folate intake (Fig 2A). There is no apparent nonlinear correlation between the intake of dietary folate and the risk of all-cause mortality (P = .298). Fig 2B shows a decreasing trend in the risk of CVD death with increasing folic acid intake (Fig 2B). However, the overall association between folate intake and risk of CVD death did not achieve statistical significance (P overall = 0.075, P nonlinear = 0.187).

**Table 1. Baseline characteristics of participants with NAFLD based on dietary folate intake levels.**

| Characteristics | Dietary folate intake (µg/d) | | | | Total | P value |
|---|---|---|---|---|---|---|
| | Quartile1 ≤250µg/d | Quartile2 250.5–342.5µg/d | Quartile3 343–467µg/d | Quartile4 ≥467.5µg/d | | |
| **Patients, n(%)** | 814 (24.9%) | 817 (25.0%) | 818 (25.0%) | 817 (25.0%) | 3,266 (100.0%) | |
| **Age(years)** | | | | | | <0.001 |
| ≤39 | 202 (24.8%) | 173 (21.2%) | 188 (23.0%) | 250 (30.6%) | 813 (24.9%) | |
| 40–59 | 231 (28.4%) | 264 (32.3%) | 278 (34.0%) | 277 (33.9%) | 1,050 (32.1%) | |
| ≥60 | 381 (46.8%) | 380 (46.5%) | 352 (43.0%) | 290 (35.5%) | 1,403 (43.0%) | |
| **GENDER** | | | | | | <0.001 |
| Male | 340 (41.8%) | 407 (49.8%) | 476 (58.2%) | 582 (71.2%) | 1,805 (55.3%) | |
| Female | 474 (58.2%) | 410 (50.2%) | 342 (41.8%) | 235 (28.8%) | 1,461 (44.7%) | |
| **RACE** | | | | | | <0.001 |
| Mexican American | 216 (26.5%) | 226 (27.7%) | 208 (25.4%) | 220 (26.9%) | 870 (26.6%) | |
| Other Hispanic | 89 (10.9%) | 87 (10.6%) | 83 (10.1%) | 66 (8.1%) | 325 (10.0%) | |
| Non-Hispanic White | 341 (41.9%) | 357 (43.7%) | 401 (49.0%) | 430 (52.6%) | 1,529 (46.8%) | |
| Non-Hispanic Black | 134 (16.5%) | 117 (14.3%) | 82 (10.0%) | 56 (6.9%) | 389 (11.9%) | |
| Other Race | 34 (4.2%) | 30 (3.7%) | 44 (5.4%) | 45 (5.5%) | 153 (4.7%) | |
| **Education level** | | | | | | <0.001 |
| Less than high school | 362 (44.5%) | 316 (38.7%) | 276 (33.7%) | 236 (28.9%) | 1,190 (36.4%) | |
| high school | 205 (25.2%) | 196 (24.0%) | 184 (22.5%) | 171 (20.9%) | 756 (23.1%) | |
| College or above | 247 (30.3%) | 305 (37.3%) | 358 (43.8%) | 410 (50.2%) | 1,320 (40.4%) | |
| **Income level** | | | | | | <0.001 |
| <1 | 213 (26.2%) | 152 (18.6%) | 148 (18.1%) | 155 (19.0%) | 668 (20.5%) | |
| 1–3 | 436 (53.6%) | 440 (53.9%) | 403 (49.3%) | 343 (42.0%) | 1,622 (49.7%) | |
| ≥3 | 165 (20.3%) | 225 (27.5%) | 267 (32.6%) | 319 (39.0%) | 976 (29.9%) | |
| **BMI(kg/m²)** | | | | | | 0.792 |
| <25 | 44 (5.4%) | 38 (4.7%) | 41 (5.0%) | 40 (4.9%) | 163 (5.0%) | |
| 25–29.9 | 217 (26.7%) | 201 (24.6%) | 223 (27.3%) | 241 (29.5%) | 882 (27.0%) | |
| 30–34.9 | 269 (33.0%) | 274 (33.5%) | 279 (34.1%) | 271 (33.2%) | 1,093 (33.5%) | |
| 35–39.9 | 154 (18.9%) | 173 (21.2%) | 150 (18.3%) | 144 (17.6%) | 621 (19.0%) | |
| ≥40 | 130 (16.0%) | 131 (16.0%) | 125 (15.3%) | 121 (14.8%) | 507 (15.5%) | |
| **Smoking status** | | | | | | 0.210 |
| Never smokers | 428 (52.6%) | 416 (50.9%) | 421 (51.5%) | 438 (53.6%) | 1,703 (52.1%) | |
| Former smokers | 230 (28.3%) | 247 (30.2%) | 261 (31.9%) | 257 (31.5%) | 995 (30.5%) | |
| Current smokers | 156 (19.2%) | 154 (18.8%) | 136 (16.6%) | 122 (14.9%) | 568 (17.4%) | |
| **Hypertension** | | | | | | 0.545 |
| Yes | 418 (51.4%) | 425 (52.0%) | 402 (49.1%) | 402 (49.2%) | 1,647 (50.4%) | |
| No | 396 (48.6%) | 392 (48.0%) | 416 (50.9%) | 415 (50.8%) | 1,619 (49.6%) | |
| **Cardiovascular disease** | | | | | | 0.002 |
| Yes | 144 (17.7%) | 119 (14.6%) | 97 (11.9%) | 98 (12.0%) | 458 (14.0%) | |
| No | 670 (82.3%) | 698 (85.4%) | 721 (88.1%) | 719 (88.0%) | 2,808 (86.0%) | |
| **CANCER** | | | | | | 0.388 |
| Yes | 90 (11.1%) | 77 (9.4%) | 95 (11.6%) | 97 (11.9%) | 359 (11.0%) | |
| No | 724 (88.9%) | 740 (90.6%) | 723 (88.4%) | 720 (88.1%) | 2,907 (89.0%) | |
| **DIABETES** | | | | | | <0.001 |
| Yes | 218 (26.8%) | 239 (29.3%) | 182 (22.2%) | 172 (21.1%) | 811 (24.8%) | |
| No | 596 (73.2%) | 578 (70.7%) | 636 (77.8%) | 645 (78.9%) | 2,455 (75.2%) | |

*(Continued)*

**Table 1.** (Continued)

| Characteristics | Dietary folate intake (µg/d) | | | | | P value |
|---|---|---|---|---|---|---|
| | Quartile1 ≤250µg/d | Quartile2 250.5–342.5µg/d | Quartile3 343–467µg/d | Quartile4 ≥467.5µg/d | Total | |
| **Dyslipidemia** | | | | | | 0.874 |
| Yes | 606 (74.4%) | 599 (73.3%) | 598 (73.1%) | 594 (72.7%) | 2,397 (73.4%) | |
| No | 208 (25.6%) | 218 (26.7%) | 220 (26.9%) | 223 (27.3%) | 869 (26.6%) | |
| **Physical activity** | | | | | | <0.001 |
| Active | 280 (34.4%) | 335 (41.0%) | 366 (44.7%) | 410 (50.2%) | 1,391 (42.6%) | |
| Inactive | 534 (65.6%) | 482 (59.0%) | 452 (55.3%) | 407 (49.8%) | 1,875 (57.4%) | |
| **Use of dietary supplement** | | | | | | 0.004 |
| Yes | 338 (41.5%) | 377 (46.1%) | 407 (49.8%) | 399 (48.8%) | 1,521 (46.6%) | |
| No | 476 (58.5%) | 440 (53.9%) | 411 (50.2%) | 418 (51.2%) | 1,745 (53.4%) | |
| **ALT,median(IQR) U/L** | 24 (18–34) | 23 (19–32) | 25 (20–34) | 28 (21–38) | 25(19–35) | <0.001 |
| **AST,median(IQR), U/L** | 24 (21–29) | 23 (20–28) | 25 (21–29.8) | 25 (21–30) | 24(21–29) | 0.025 |
| **HDL-C, median (IQR), mg/dL** | 45 (38–53) | 45 (39–53) | 44 (38–52) | 43 (38–50) | 44(38–52) | <0.001 |
| **LDL, median (IQR), mg/dL** | 113 (92–142) | 115 (90–139) | 113 (91–139) | 113 (91–136) | 113.6(91–139) | 0.198 |
| **TC, median (IQR), mg/dL** | 195(168–222) | 194 (167–224) | 193 (167–223) | 189 (163–218) | 193 (166–222) | 0.038 |
| **TG, median (IQR), mg/dL** | 140 (99–196) | 141 (101–202) | 147 (103–200) | 145 (100–207) | 143 (101–202) | 0.988 |
| **Castelli_II index** | | | | | | 0.051 |
| <3 | 517 (63.5%) | 562 (68.8%) | 559 (68.3%) | 527 (64.5%) | 2,165 (66.3%) | |
| ≥3 | 297 (36.5%) | 255 (31.2%) | 259 (31.7%) | 290 (35.5%) | 1,101 (33.7%) | |
| **Use of methotrexate** | | | | | | 0.718 |
| Yes | 4 (0.5%) | 4 (0.5%) | 2 (0.2%) | 2 (0.2%) | 12 (0.4%) | |
| No | 810 (99.5%) | 813 (99.5%) | 816 (99.8%) | 815 (99.8%) | 3,254 (99.6%) | |
| **Use of methotrexate** | | | | | | 0.905 |
| Yes | 103 (12.7%) | 99 (12.1%) | 94 (11.5%) | 101 (12.4%) | 397 (12.2%) | |
| No | 711 (87.3%) | 718 (87.9%) | 724 (88.5%) | 716 (87.6%) | 2,869 (87.8%) | |
| **Daily sugar intake(g/d)** | | | | | | <0.001 |
| ≤50 | 259 (31.8%) | 129 (15.8%) | 75 (9.2%) | 48 (5.9%) | 511 (15.6%) | |
| >50 | 555 (68.2%) | 688 (84.2%) | 743 (90.8%) | 769 (94.1%) | 2,755 (84.4%) | |

Abbreviations: ALT, alanine aminotransferase; AST, aspartate aminotransferase; BMI, body mass index (calculated as weight in kilograms divided by height in meters squared); HDL-C, high-density lipoprotein cholesterol; LDL-C, low-density lipoprotein cholesterol; NAFLD, non-alcoholic fatty liver disease; TC, total cholesterol; TG, triglyceride.

## Stratified analysis and sensitivity analysis

There was a significant correlation (p = 0.007) found between the risk of all-cause mortality and the dietary folate intake of different racial groups. Mexican Americans in the highest quartile had a lower HR of 0.26 (95% CI, 0.11–0.61), while Other Hispanics had an increased HR of 4.41 (95% CI, 1.19–16.39) in the same quartile. Participants with a daily sugar intake >50 g/d had a lower mortality risk in the highest folate quartile (HR, 0.56; 95% CI, 0.40–0.78), whereas no significant reduction was seen in those with sugar intake ≤50 g/d (HR, 1.99; 95% CI, 0.91–4.32). There were no significant interactions between race, gender, age, education levels, BMI, income bracket, smoking habits, presence of diabetes, hypertension, history of cardiovascular disease, dyslipidemia, and level of physical activity with dietary folate intake in relation to all-cause mortality (Table 3).

**Table 2. Hazard ratios for all-cause and CVD mortality among participants with NAFLD.**

| Model | Dietary folate intake (μg/d) | | | | Trend p-value |
|---|---|---|---|---|---|
| | Quartile1 ≤250μg/d | Quartile2 250.5–342.5μg/d | Quartile3 343–467μg/d | Quartile4 ≥467.5μg/d | |
| All-cause mortality | | | | | |
| Model 1 | 1.00 | 0.79 (0.60–1.03) | 0.73 (0.56–0.96) | 0.57 (0.44–0.75) | <0.001 |
| Model 2 | 1.00 | 0.70 (0.54–0.92) | 0.60 (0.46–0.79) | 0.53 (0.40–0.71) | <0.001 |
| Model 3 | 1.00 | 0.78 (0.59–1.03) | 0.77 (0.58–1.02) | 0.69 (0.51–0.94) | 0.028 |
| CVD mortality | | | | | |
| Model 1 | 1.00 | 0.73 (0.46–1.15) | 0.66 (0.42–1.04) | 0.39 (0.24–0.64) | <0.001 |
| Model 2 | 1.00 | 0.68 (0.42–1.09) | 0.58 (0.36–0.96) | 0.41 (0.23–0.70) | 0.001 |
| Model 3 | 1.00 | 0.74 (0.44–1.27) | 0.80 (0.45–1.41) | 0.55 (0.29–1.04) | 0.107 |

Abbreviations: CVD, cardiovascular disease; NAFLD, non-alcoholic fatty liver disease.

Model 1:no covariates were adjusted; Model 2: adjusted sex, age, and race/ethnicity; Model 3: model 2 plus education level, income level, BMI, hypertension, cardiovascular disease, cancer, physical activity, use of dietary supplements, diabetes, smoking status, dyslipidemia, alt, TC, Castelli_II index, use of methotrexate, use of methotrexate, daily sugar intake.

In sensitivity analyses, when examining dietary folate consumption as a continuous variable, a consistent negative connection with all-cause mortality is shown in Models 1, 2, and 3 (S1 Table in S1 File), models 1 and 2 also demonstrate a negative correlation with CVD mortality rates (S2 Table in S1 File). After excluding patients who died within the two-year follow-up period, the relationship between dietary folate intake and both all-cause mortality and cardiovascular mortality remained robust in Models 1 and 2 (S3 and S4 Tables in S1 File). The results continued to be stable after excluding people who only had a single recollection of dietary folate intake (S5 and S6 Tables in S1 File). Furthermore, after removing non-alcoholic fatty liver disease patients who passed away within two years of follow-up and had just one dietary recall, the relationship between the intake of dietary folate and both all-cause and CVD mortality rates did not undergo any substantial changes in Models 1 and 2 (S7 and S8 Tables in S1 File).

## Discussion

This cohort study aimed to examine the correlation between dietary intake of folate and both mortality of all cause and CVD mortality in patients with NAFLD through a comprehensive analysis of a large group of participants. After adjusting for various potential confounding factors, we discovered that higher dietary folate intake was linked to a reduced risk of all-cause mortality in NAFLD patients, but no significant association was found between higher dietary folate intake and a lower risk of CVD mortality. Our study's solidity has been affirmed through the application of both layered scrutiny and susceptibility examination.

This is the first study, as far as we know, to evaluate the relationship between the total mortality and CVD mortality in people with NAFLD and the dietary folate intake. Prior studies have investigated the link between dietary folate intake, serum folic acid, and NAFLD. In a study utilizing data from the NHANES 2007–2014, Bo Zhang and colleagues examined the relationship between dietary folic acid intake and NAFLD. Their findings indicated a negative correlation between the intake of dietary folate and the risk of NAFLD among the American population [10]. A cross-sectional investigation was carried out by Yushi Chen and his team on 7,543 adults, utilizing data from NHANES 2009–2018. The findings indicate that fewer American adults with NAFLD have higher serum and dietary folate levels [23]. In a cross-

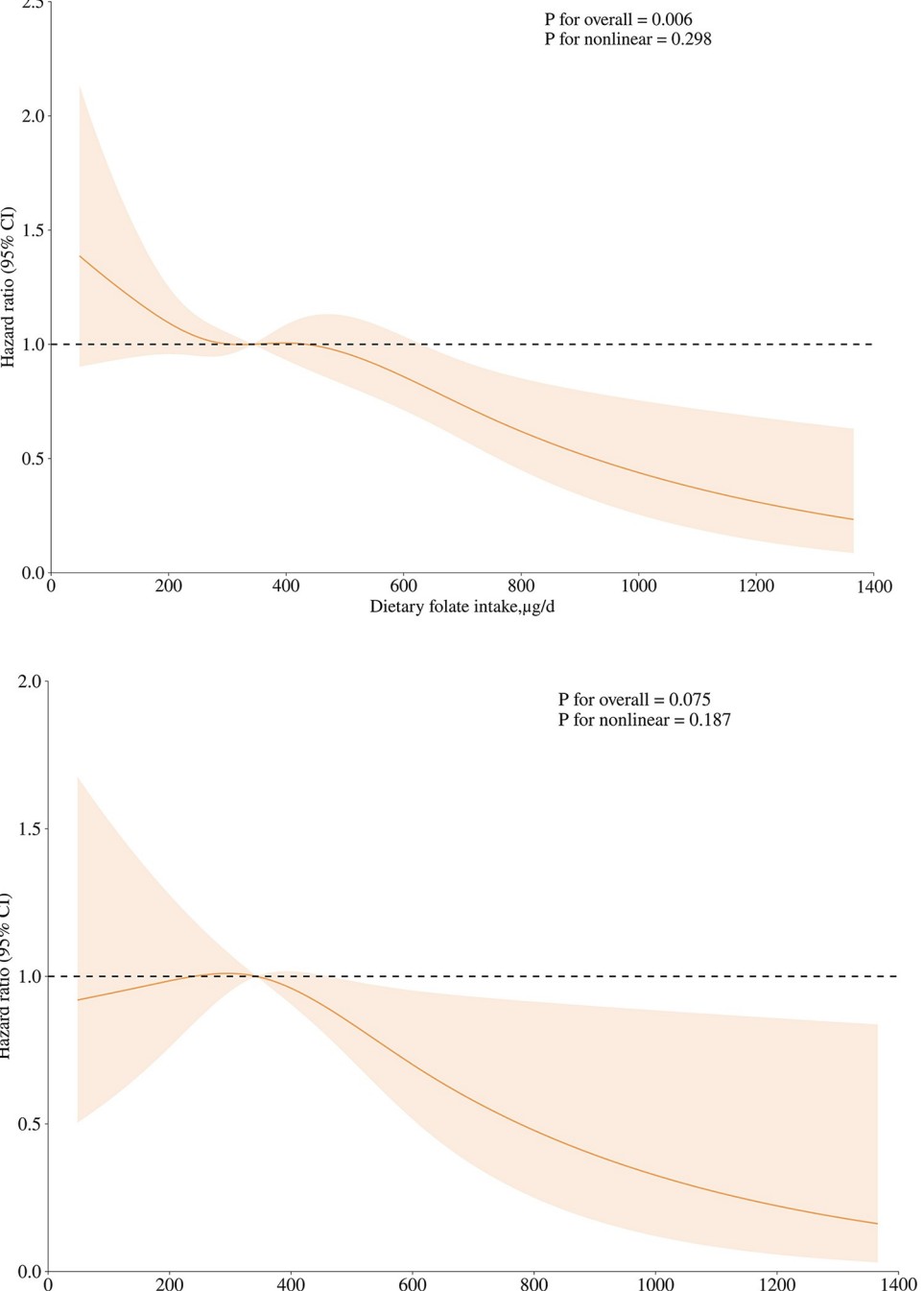

**Fig 2. Association of dietary folate intake with all-cause mortality and CVD mortality among individuals with NAFLD.** A: Association of Dietary Folate Intake with All-Cause Mortality; B: Association of Dietary Folate Intake with CVD Mortality. Note: Hazard ratios were adjusted for sex, age, race/ethnicity, education level, income level, BMI, hypertension, cardiovascular disease, cancer, physical activity, use of dietary supplements, diabetes, smoking status, dyslipidemia, alanine aminotransferase, total cholesterol, Castelli_II index, use of methotrexate, use of methotrexate, daily sugar intake. NAFLD, nonalcoholic fatty liver disease. Shaded areas represent 95% of CIs.

**Table 3. Associations of dietary folate intake with all-cause mortality in different subgroups among participants with NAFLD.**

| Characteristic | Dietary folate intake (µg/d) | | | | |
|---|---|---|---|---|---|
| | Quartile1 ≤250µg/d | Quartile2 250.5–342.5µg/d | Quartile3 343–467µg/d | Quartile4 ≥467.5µg/d | p for interaction |
| **Age(years)** | | | | | 0.936 |
| ≤39 | 1.00 | 1.06 (0.13–9.02) | 1.15 (0.29–4.63) | 0.57 (0.08–3.88) | |
| 40–59 | 1.00 | 0.62 (0.26–1.46) | 0.81 (0.38–1.70) | 0.72 (0.32–1.59) | |
| ≥60 | 1.00 | 0.83 (0.61–1.12) | 0.78 (0.57–1.06) | 0.69 (0.49–0.99) | |
| **GENDER** | | | | | 0.427 |
| Male | 1.00 | 0.78 (0.53–1.15) | 0.68 (0.46–1.00) | 0.58 (0.39–0.86) | |
| Female | 1.00 | 0.74 (0.48–1.13) | 0.81 (0.52–1.27) | 1.03 (0.63–1.69) | |
| **RACE** | | | | | 0.012 |
| Mexican American | 1.00 | 0.71 (0.38–1.33) | 0.70 (0.35–1.40) | 0.22 (0.09–0.54) | |
| Other Hispanic | 1.00 | 1.39 (0.43–4.44) | 0.87 (0.15–4.87) | 2.62 (0.65–10.60) | |
| Non-Hispanic White | 1.00 | 0.76 (0.54–1.06) | 0.70 (0.50–0.98) | 0.67 (0.47–0.96) | |
| Non-Hispanic Black | 1.00 | 0.93 (0.44–1.96) | 1.67 (0.69–4.01) | 1.35 (0.50–3.61) | |
| Other Race | 1.00 | 3.45 (0.08–155.72) | 1.04 (0.07–14.55) | 0.85 (0.08–8.81) | |
| **Education level** | | | | | 0.085 |
| Less than high school | 1.00 | 0.74 (0.51–1.07) | 1.08 (0.72–1.62) | 0.86 (0.53–1.38) | |
| high school | 1.00 | 1.12 (0.65–1.92) | 0.63 (0.36–1.12) | 0.64 (0.32–1.30) | |
| College or above | 1.00 | 0.59 (0.35–0.99) | 0.59 (0.34–1.02) | 0.57 (0.35–0.93) | |
| **Income level** | | | | | 0.486 |
| <1 | 1.00 | 0.73 (0.38–1.40) | 0.83 (0.44–1.55) | 0.41 (0.19–0.87) | |
| 1–3 | 1.00 | 0.74 (0.52–1.05) | 0.83 (0.57–1.19) | 0.62 (0.40–0.95) | |
| ≥3 | 1.00 | 0.71 (0.35–1.43) | 0.55 (0.28–1.05) | 0.63 (0.32–1.23) | |
| **Smoking status** | | | | | 0.684 |
| Never smokers | 1.00 | 1.04 (0.67–1.63) | 0.91 (0.55–1.50) | 0.75 (0.45–1.27) | |
| Former smokers | 1.00 | 0.57 (0.38–0.84) | 0.53 (0.36–0.79) | 0.53 (0.34–0.81) | |
| Current smokers | 1.00 | 0.61 (0.31–1.21) | 0.91 (0.45–1.85) | 0.65 (0.31–1.37) | |
| **BMI(kg/m2)** | | | | | 0.385 |
| <25 | 1.00 | 0.34 (0.13–0.91) | 0.91 (0.37–2.19) | 0.41 (0.15–1.14) | |
| 25–29.9 | 1.00 | 0.96 (0.56–1.63) | 0.73 (0.43–1.22) | 0.77 (0.43–1.36) | |
| 30–34.9 | 1.00 | 1.05 (0.65–1.70) | 0.79 (0.49–1.26) | 0.60 (0.35–1.03) | |
| 35–39.9 | 1.00 | 0.40 (0.19–0.88) | 0.70 (0.30–1.61) | 0.82 (0.34–2.00) | |
| ≥40 | 1.00 | 0.78 (0.33–1.81) | 1.15 (0.49–2.73) | 0.66 (0.25–1.78) | |
| **DIABETES** | | | | | 0.065 |
| Yes | 1.00 | 0.53 (0.34–0.83) | 0.82 (0.51–1.32) | 0.70 (0.42–1.15) | |
| No | 1.00 | 0.99 (0.68–1.43) | 0.73 (0.50–1.07) | 0.71 (0.48–1.06) | |
| **Hypertension** | | | | | 0.497 |
| Yes | 1.00 | 0.81 (0.57–1.14) | 0.85 (0.59–1.21) | 0.79 (0.54–1.14) | |
| No | 1.00 | 0.78 (0.47–1.30) | 0.61 (0.38–0.99) | 0.62 (0.36–1.05) | |
| **Cardiovascular disease** | | | | | 0.944 |
| Yes | 1.00 | 0.66 (0.41–1.06) | 0.70 (0.43–1.14) | 0.63 (0.36–1.08) | |
| No | 1.00 | 0.86 (0.60–1.22) | 0.79 (0.56–1.13) | 0.75 (0.51–1.11) | |
| **Dyslipidemia** | | | | | 0.367 |
| Yes | 1.00 | 0.67 (0.47–0.94) | 0.72 (0.51–1.02) | 0.66 (0.46–0.96) | |
| No | 1.00 | 1.17 (0.73–1.88) | 0.98 (0.59–1.63) | 0.84 (0.47–1.52) | |
| **Physical activity** | | | | | 0.911 |
| active | 1.00 | 0.81 (0.58–1.13) | 0.82 (0.58–1.14) | 0.68 (0.47–0.99) | |

*(Continued)*

**Table 3.** (Continued)

| Characteristic | Dietary folate intake (µg/d) | | | | |
|---|---|---|---|---|---|
| | Quartile1 ≤250µg/d | Quartile2 250.5–342.5µg/d | Quartile3 343–467µg/d | Quartile4 ≥467.5µg/d | p for interaction |
| inactive | 1.00 | 0.75 (0.44–1.28) | 0.71 (0.42–1.20) | 0.78 (0.44–1.38) | |
| **Daily sugar intake(g/d)** | | | | | 0.002 |
| ≤50 | 1.00 | 1.07 (0.61–1.86) | 0.83 (0.44–1.56) | 1.99 (0.91–4.32) | |
| >50 | 1.00 | 0.69 (0.50–0.94) | 0.69 (0.50–0.94) | 0.56 (0.40–0.78) | |

Abbreviations: BMI, body mass index (calculated as weight in kilograms divided by height in meters squared); NAFLD, non-alcoholic fatty liver disease. Adjusted for sex, age, race/ethnicity, education level, income level, BMI, hypertension, cardiovascular disease, cancer, physical activity, use of dietary supplements, diabetes, smoking status, dyslipidemia, alt, TC, Castelli_II index, use of methotrexate, use of methotrexate, daily sugar intake; The stratum variable was excluded while performing stratification solely based on it.

sectional analysis by Hao-Kai Chen and his team, which included 5,417 individuals and utilized data from the NHANES 2011–2018, it was discovered that higher levels of serum folate were linked to a reduced likelihood of NAFLD among adults in the United States [9]. In a randomized controlled trial, Anat Yaskolka Meir and colleagues studied 294 participants in Israel with abdominal obesity or lipid abnormalities. They found that an increase in serum folic acid was associated with a greater reduction in intrahepatic fat (IHF) among the participants, suggesting that higher serum folic acid levels may be related to a decreased risk of NAFLD [24]. In a case-control study conducted by Hamid Zolfaghari and team in Isfahan, involving 317 subjects, it was noted that people with NAFLD consumed less folate on average in their diets than did healthy people [25]. Nevertheless, a cross-sectional study undertaken by Xiaohui Liu and colleagues, which included 4,352 people, found no correlation between dietary folic acid intake and NAFLD [26]. Serum folate levels and NAFLD were not correlated in a case-control study by Li Li and colleagues with 8,397 participants [27]. In a study spearheaded by Jung Mi Han in South Korea, which involved 348 subjects, the findings suggested that there was no significant link between the intake of folate and the occurrence of NAFLD [28]. The conflicting results on the association between folate consumption and NAFLD may arise from differences in dietary patterns or approaches to assessing dietary intake among diverse groups.

A cohort study of 14,234 people at high cardiovascular risk by Xiaoqing Xu et al. revealed that a moderate intake of folic acid was linked to a lower risk of all-cause and CVD mortality, whereas a higher intake did not consistently lower these risks [29]. Additional research has investigated the correlation between levels of folate in the blood and the risk of mortality. Yang Peng and others used 1999–2010 NHANES data to conduct a cohort study on 28,854 subjects, the results showed that lower folate levels are linked to higher all-cause mortality and CVD mortality [30]. A prospective cohort study by Yujie Liu and colleagues including 8,067 T2DM patients revealed a higher risk of CVD death in T2DM patients with low serum folate levels [31]. A cohort investigation was carried out by Kalyani Sonawane and her team on 683 individuals suffering from rheumatoid arthritis. The findings indicate a correlation between elevated folate levels in the blood and a reduced risk of death from CVD among these patients [32]. Another cohort study was conducted by Stanley Nkemjika and his colleagues on 3,116 participants using data from NHANES 1991–1994. The study revealed that among adults with hypertension, those with either low or high folate levels, as compared to those with moderate levels of serum folate, have a heightened risk of dying from cardiovascular diseases [33]. In this cohort study, we observed the benefits of higher dietary folate intake in 3,266 American adult NAFLD patients, specifically, higher dietary folate intake (≥467.5µg/d vs ≤250µg/d /day) is

linked to a 31% reduction in the risk of all-cause mortality in American adults with NAFLD. Certain results imply that a higher folate intake or serum folate levels may be linked to a higher risk of death, and this difference may be explained by the various disease types of the participants chosen for the study.

The National Institutes of Health (NIH) advises adult consumption of 400μg of folate per day. This recommendation is based on the role of folate in DNA synthesis, repair, and methylation, as well as its importance in preventing neural tube defects during early pregnancy [34]. The results in Table 2 and Fig 2A of this study indicate that the HR for all-cause mortality was less than 1 when dietary folic acid intake exceeded467.5μg/d. Therefore, a dietary folic acid intake of more than467.5μg/d may be a significant protective factor for people with NAFLD. Putting together the data from Table 2 and Fig 2B, it is evident that the highest intake group (≥467.5μg/d) had a much-reduced risk of CVD death, and while there might have been notable variations at particular intake levels, the overall trend was not significant. To ascertain the precise mechanism of this association, more research is therefore required.

A Systematic Review and Meta-analysis were conducted by Nicole E Rich and her team on the American populace, the data suggest a significant discrepancy in the occurrence and seriousness of NAFLD among various racial and ethnic groups in the U.S., with the highest rates observed among Hispanics and the lowest among African Americans [35]. The variation in NAFLD prevalence across different races could be attributed to distinct dietary practices and genetic predispositions to NAFLD. This could, in part, elucidate the observed differences in the correlation between dietary folate consumption and the risk of all-cause mortality among various racial groups in this research [36].

Recent evidence indicates that excessive sugar consumption, particularly fructose, accelerates hepatic steatosis, contributing to the onset and progression of NAFLD [37]. In a cohort study of 98,786 postmenopausal women, Zhao et al. found that chronic liver disease mortality was significantly higher among those consuming one or more servings of sugar-sweetened beverages daily compared to those with lower consumption (three or fewer servings per month) [38]. Combined with the findings from Table 3, a higher folate intake appeared to offer protective effects against all-cause mortality in individuals with moderate to high sugar consumption. The absence of statistical significance in those with low sugar intake may be attributed to the limited sample size in this subgroup. While the underlying process linking dietary folate consumption and the risk of death remains uncertain, several interpretations exist. One such interpretation is that a lack of folate can result in increased homocysteine levels, which in turn can stimulate the build-up of fat, thereby causing NAFLD and atherosclerotic changes [39–41]. Furthermore, hyperhomocysteinemia is viewed as a contributing factor to the development and mortality of cardiovascular diseases [42–44]. Alternate research suggests that folate can mitigate oxidative stress by eliminating reactive oxygen species, which in turn, through an inflammatory response, can accelerate the development and advancement of NAFLD [45–47]. Furthermore, an increase in folate intake can stimulate AMPK, and the reestablishment of AMPK functionality could potentially enhance the metabolism of glucose and cholesterol, which is often compromised due to the intake of a high-fat diet [48]. This could subsequently suppress the production of liver fat and ameliorate hepatic steatosis [49]. Moreover, a lack of folate can disrupt the methylation of DNA, which in turn can influence the process of human aging and exert an effect on the rate of mortality [50].

This research underscores the potential advantages of higher dietary folate intake in reducing overall mortality among American adults with NAFLD. Healthcare providers might consider folate as a modifiable dietary factor that could enhance long-term outcomes for this patient population. Although a significant reduction in cardiovascular disease mortality was not observed, the general trend toward lower all-cause mortality indicates that dietary

recommendations promoting increased folate consumption may offer a beneficial strategy for improving prognosis in NAFLD patients. Incorporating adequate folate intake into dietary guidelines and management plans could help mitigate the associated risks of NAFLD. However, additional studies are necessary to clarify the mechanisms by which folate influences mortality and to determine the ideal intake levels for different patient subgroups.

## Strengths and limitations

Our research has some advantages. First, this study utilized data from NHANES, which is representative of the U.S. population. Second, the large sample size, rigorous follow-up procedures, and careful adjustment for confounding variables increase the confidence in the results. Third, to the best of our knowledge, there are no studies on the relationship between dietary folate intake and mortality risk. The novelty of this study lies in its comprehensive approach, integrating years of robust data from NHANES and using advanced statistical methods such as Cox proportional risk regression models and restricted cubic spline. The use of these statistical methods allowed for a careful analysis of these associations, providing valuable insights into the dose-response relationship between dietary folate intake and mortality risk.

Our study also has some limitations. First, While the WWEIA tool provides reliable data on short-term dietary intake, it may not fully capture long-term eating habits. The two 24-hour recalls used in this study reflect current consumption, which can vary from day to day. This limitation should be considered when interpreting the relationship between folate intake and mortality. Future studies could benefit from more comprehensive tools to assess habitual dietary patterns. Second, although we have accounted for potential confounders such as physical activity and diabetes, there may be other unmeasured or residual confounding factors that could influence our findings. Third, although the NHANES data are representative of the U.S. population, our study population may not be representative of all individuals with NAFLD, limiting the generalizability of our findings. Fourth, due to the limitations of the database, we were unable to account for dietary patterns such as omnivorous or vegetarian diets, which may have led to some deviation of the endings from reality. In addition, dietary folate intake is a lifestyle behavior that may change over time and have an impact on study outcomes. Lastly, further investigation is needed to show a cause-and-effect association between dietary folate intake and mortality in patients with NAFLD, despite the study being a cohort study. Future longitudinal studies with larger and more diverse samples, objective measures of dietary intake and folate levels, and comprehensive adjustment for potential confounders are needed to confirm and extend our findings.

## Conclusions

The findings from our cohort investigation, focusing on adult individuals with NAFLD in the United States, indicate a potential correlation between increased intake of dietary folate and a reduced likelihood of all-cause mortality. In this study, a higher intake of dietary folate was not linked to a reduced risk of CVD mortality. Our findings contribute to dietary guidance for NAFLD patients and improve their prognosis. The correlation between dietary folate consumption and the reduction in overall and cardiovascular disease-related mortality risk among NAFLD patients, particularly in terms of dose response, warrants further exploration to establish the ideal intake quantities.

## Supporting information

**S1 File.**
(DOCX)

**S1 Raw data.**
(XLSX)

## Acknowledgments

The authors express gratitude to the National Center for Health Statistics of the Centers for Disease Control and Prevention for their collaboration in providing the data.

## Author Contributions

**Conceptualization:** Jinsheng Dong, Runshun Zhang, Yuning Bai.

**Data curation:** Jinsheng Dong, Zhiqiang Li, Chenlu Wang.

**Funding acquisition:** Yuning Bai, Wenliang Lv.

**Investigation:** Jinsheng Dong.

**Methodology:** Jinsheng Dong, Zhiqiang Li, Runshun Zhang, Mingkun Liu.

**Project administration:** Chenlu Wang, Yilin Li, Yuning Bai, Wenliang Lv.

**Software:** Jinsheng Dong.

**Supervision:** Zhiqiang Li, Chenlu Wang, Runshun Zhang, Yilin Li, Mingkun Liu, Qiuye Chen, Yuning Bai, Wenliang Lv.

**Validation:** Jinsheng Dong, Zhiqiang Li, Qiuye Chen.

**Writing – original draft:** Jinsheng Dong, Yuning Bai.

**Writing – review & editing:** Jinsheng Dong, Zhiqiang Li, Chenlu Wang, Runshun Zhang, Yilin Li, Mingkun Liu, Qiuye Chen, Yuning Bai, Wenliang Lv.

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
