## [Decision Letter · Decision Letter 0]

11 Sep 2024

PONE-D-24-27398Dietary folate intake and all-cause mortality and cardiovascular mortality in American adults with non-alcoholic fatty liver disease: Data from NHANES 2003 to 2018PLOS ONE

Dear Dr. Bai,

Thank you for submitting your manuscript to PLOS ONE. After careful consideration, we feel that it has merit but does not fully meet PLOS ONE’s publication criteria as it currently stands. Therefore, we invite you to submit a revised version of the manuscript that addresses the points raised during the review process.

We look forward to receiving your revised manuscript.

Kind regards,

Aleksandra Klisic

Academic Editor

PLOS ONE

“This study was supported by the Scientific and Technological Innovation Project of China Acad-emy of Chinese Medical Sciences(no.CI2021A00806).”

3. We note that there is identifying data in the Supporting Information file < rawdata.xlsx>. Due to the inclusion of these potentially identifying data, we have removed this file from your file inventory. Prior to sharing human research participant data, authors should consult with an ethics committee to ensure data are shared in accordance with participant consent and all applicable local laws.

-Location data

Reviewers' comments:

Reviewer's Responses to Questions

**Comments to the Author**

1. Is the manuscript technically sound, and do the data support the conclusions?

Reviewer #1: Partly

Reviewer #2: Yes

2. Has the statistical analysis been performed appropriately and rigorously? 

Reviewer #1: Yes

Reviewer #2: Yes

3. Have the authors made all data underlying the findings in their manuscript fully available?

Reviewer #1: Yes

Reviewer #2: No

4. Is the manuscript presented in an intelligible fashion and written in standard English?

Reviewer #1: Yes

Reviewer #2: Yes

5. Review Comments to the Author

Reviewer #1: The present work deals with the evaluation of the NHANES database from 2003 to 2018, which sought to evaluate the correlation between dietary folate consumption and mortality from all causes and cardiovascular diseases. The premise of the authors' hypothesis is interesting and, in fact, innovative. The statistical method used by the authors is appropriate, as is the way of reporting in the scientific article. However, some points concern me with the conclusions generated by the authors. Therefore, I leave some comments and suggestions that aim to reinforce the quality of the article, as analyzes for confounding variables were explored, but little was explored regarding nutritional determinants:

• I suggest adding a covariate to analyze folate for autoimmune diseases, especially for NAFLD patients using methotrexate or NAFLD patients using metformin. Both medications impair folate metabolism.

• I suggest subdividing the BMI ≥ 30.0 g/m² category (obesity) into the three subclasses of obesity, grade I, grade II, and grade III. This may provide more information about the severity or relationship of body components to daily folate intake.

• I recommend adding information on biochemical analytes to the Castelli II Index, which uses the ratio of LDL-c to HDL-c. Authors can use categories of high cardiovascular risk (above the cutoff point) or no cardiovascular risk (below the cutoff point). Additionally, the authors could use Framingham Score with this database to further separate heart risk. This can provide more information than just isolated lipoproteins.

• The authors use WWEIA as a tool to separate daily dietary folate intake into quartiles. The tool itself is suitable for what it measures: current food consumption. The two 24-hour dietary recalls do not measure eating habits. Therefore, I believe it is important to emphasize more on the way it is written in the conclusion and discussion in the current version of the manuscript.

• I believe it is important to evaluate, if possible, the participants in this sample between omnivores and vegans. The dietary pattern itself can be a confounding factor.

• As an additional analysis to evaluate model 3, I strongly recommend extracting information from the database for total daily sugar intake. The focus would be to evaluate consumption below 50g of sugar per day, compared to participants who exceed this daily value.

Reviewer #2: Dong et al. have performed a study on dietary folate intake with all-cause mortality and CV mortality in adults with NAFLD using NHANES data. The manuscript is well-written and the findings are interesting. These are my comments:

- A paragraph summarizing the clinical take-home message of this manuscript should be added to the discussion.

- The references prior to 2010 could be updated with those after 2010 since they provide more up-to-date findings.

6. PLOS authors have the option to publish the peer review history of their article (what does this mean?). If published, this will include your full peer review and any attached files.

Reviewer #1: **Yes: **Giuseppe Potrick Stefani

Reviewer #2: No

---

## [Author Response · Author response to Decision Letter 0]

21 Oct 2024

Dear Editor and Reviewers:

We appreciate the opportunity to allow us to revise our manuscript and thanks for reviewers’ constructive comments and suggestions. We would like to submit our revised manuscript, entitled “Dietary folate intake and all-cause mortality and cardiovascular mortality in American adults with non-alcoholic fatty liver disease: Data from NHANES 2003 to 2018” for consideration for publication. In the revised manuscript, we have carefully addressed all comments and questions raised by reviewers’ point-by-point. We greatly appreciate your time and efforts to improve our manuscript for publication.

Reply to Editor

1. Thank you for stating the following financial disclosure:“This study was supported by the Scientific and Technological Innovation Project of China Acad-emy of Chinese Medical Sciences(no.CI2021A00806).”Please state what role the funders took in the study. If the funders had no role, please state: "The funders had no role in study design, data collection and analysis, decision to publish, or preparation of the manuscript."If this statement is not correct you must amend it as needed.Please include this amended Role of Funder statement in your cover letter; we will change the online submission form on your behalf.

Reply：Thank you very much for your suggestion, it has been added to the FUNDING section of the manuscript: “The funders had no role in study design, data collection and analysis, decision to publish, or preparation of the manuscript.”

2. We note that there is identifying data in the Supporting Information file < rawdata.xlsx>. Due to the inclusion of these potentially identifying data, we have removed this file from your file inventory. Prior to sharing human research participant data, authors should consult with an ethics committee to ensure data are shared in accordance with participant consent and all applicable local laws.

Reply：Thanks for your suggestion, we have removed the seqn column in <rawdata.xlsx> and replaced it with the ID column with numbers 1 through 3266.

Reply to reviewers

Reviewer #1:

1. I suggest adding a covariate to analyze folate for autoimmune diseases, especially for NAFLD patients using methotrexate or NAFLD patients using metformin. Both medications impair folate metabolism.

Reply：Thank you for your thorough review and valuable suggestions. In response, we have added the usage of metformin and methotrexate to Model 3 and made the corresponding adjustments in the Covariates Determination, Statistical Analysis, Table 1, Table 2, and Figure 2A and 2B sections. Specifically, we have added the statement:

“Methotrexate and metformin use may affect folate metabolism, and their use comes from prescription medications questionnaires.”

However, due to a significant amount of missing data on autoimmune diseases, such as rheumatoid arthritis, in the current database, including this variable would introduce bias into the results. We acknowledge the importance of considering autoimmune diseases in relation to folate metabolism, and we plan to incorporate this variable in future studies when more comprehensive data is available.

2. I suggest subdividing the BMI ≥ 30.0 g/m² category (obesity) into the three subclasses of obesity, grade I, grade II, and grade III. This may provide more information about the severity or relationship of body components to daily folate intake.

Reply：Thank you for your thorough review and valuable suggestions. In response, we have subdivided the BMI ≥ 30.0 g/m² category into three subcategories: 30-34.9, 35-39.9, and 40 and above. Together with the two previous categories, this now forms five distinct BMI groups. Additionally, we have updated the Covariates Determination, Table 1, and Figure 2A and 2B sections accordingly.

3. I recommend adding information on biochemical analytes to the Castelli II Index, which uses the ratio of LDL-c to HDL-c. Authors can use categories of high cardiovascular risk (above the cutoff point) or no cardiovascular risk (below the cutoff point). Additionally, the authors could use Framingham Score with this database to further separate heart risk. This can provide more information than just isolated lipoproteins.

Reply：Thank you for your valuable suggestion. We have incorporated the Castelli II Index into Model 3 and made corresponding revisions to the Covariates Determination, Statistical Analysis, Table 1, Table 2, and Figures 2A and 2B. Specifically, we have added the following explanation in the Covariates Determination section: 

"Castelli II index is the ratio of LDL-c to HDL-c, which better reflects the risk of cardiovascular disease (less than 3 is considered a low risk of cardiovascular disease) [21]."

Regarding the use of the Framingham Score, we acknowledge its potential utility. However, since our dataset already includes the variable "Cardiovascular disease history," adding the Framingham Score at this stage could introduce collinearity. We will consider its application in future research.

4. The authors use WWEIA as a tool to separate daily dietary folate intake into quartiles. The tool itself is suitable for what it measures: current food consumption. The two 24-hour dietary recalls do not measure eating habits. Therefore, I believe it is important to emphasize more on the way it is written in the conclusion and discussion in the current version of the manuscript.

Reply：Thank you for your insightful suggestion regarding the limitations of the WWEIA tool. We have incorporated a discussion of this limitation in the Strengths and Limitations section of the manuscript:

“While the WWEIA tool provides reliable data on short-term dietary intake, it may not fully capture long-term eating habits. The two 24-hour recalls used in this study reflect current consumption, which can vary from day to day. This limitation should be considered when interpreting the relationship between folate intake and mortality. Future studies could benefit from more comprehensive tools to assess habitual dietary patterns.”

5. I believe it is important to evaluate, if possible, the participants in this sample between omnivores and vegans. The dietary pattern itself can be a confounding factor.

Reply：Thank you for your thorough review and valuable suggestions. We greatly appreciate your input. However, due to the lack of specific dietary pattern data in the database, and the fact that the questionnaire on vegetarianism is only available for the 2007-2008 and 2009-2010 cycles, including this variable would lead to a significant amount of missing data, potentially introducing bias into the results. Therefore, we are unable to include this dietary pattern as a covariate in the current study. We will certainly take your suggestion into account in future prospective studies. We have included this point in the Limitations section of our manuscript to address the potential bias related to dietary patterns, as you suggested.

“Fourth, due to the limitations of the database, we were unable to account for dietary patterns such as omnivorous or vegetarian diets, which may have led to some deviation of the endings from reality.”

6. As an additional analysis to evaluate model 3, I strongly recommend extracting information from the database for total daily sugar intake. The focus would be to evaluate consumption below 50g of sugar per day, compared to participants who exceed this daily value.

Reply：Thank you for your valuable suggestion. We have integrated daily sugar intake into Model 3 and made several corresponding revisions across the manuscript. Specifically, we categorized daily sugar intake based on WHO recommendations as low (≤50 g/day) and medium to high (>50 g/day) and incorporated this into the Covariates Determination section: "Daily sugar intake is from the same source as folate intake and is categorized as low (≤50 g/day) and medium and high (>50 g/day) according to WHO recommendations [22]."

As a result of this inclusion, we observed that the relationship between dietary folate intake and CVD mortality lost statistical significance, though the relationship between dietary folate intake and all-cause mortality remained stable. We updated the Results section of the abstract to reflect these findings: "Compared to participants with a folate intake in Quartile 1 (≤250 μg/d), those in Quartile 4 (≥467.5 μg/d) had multivariable-adjusted hazard ratios of 0.69 (95% CI, 0.51-0.94) for all-cause mortality (p for trend = 0.028) and 0.55 (95% CI, 0.29-1.04) for CVD mortality (p for trend = 0.107)." Additionally, the Conclusion section was updated to state: "Higher dietary folate intake not found to be associated with lower CVD mortality."

In the full manuscript, we also revised the Conclusion: "In this study, a higher intake of dietary folate was not linked to a reduced risk of CVD mortality." In addition, we updated Statistical Analysis, Table 1, Table 2, and Figures 2A and 2B accordingly.

In the stratified analysis, the p for interaction for sugar intake was 0.002, which indicated a significant interaction. We added the following result to the Stratified and Sensitivity Analysis section: 

"Participants with a daily sugar intake >50 g/d had a lower mortality risk in the highest folate quartile (HR, 0.56; 95% CI, 0.40–0.78), whereas no significant reduction was seen in those with sugar intake ≤50 g/d (HR, 1.99; 95% CI, 0.91–4.32)."

In the Discussion, we included additional context to explain these findings:

 "Recent evidence indicates that excessive sugar consumption, particularly fructose, accelerates hepatic steatosis, contributing to the onset and progression of NAFLD [37]. In a cohort study of 98,786 postmenopausal women, Zhao et al. found that chronic liver disease mortality was significantly higher among those consuming one or more servings of sugar-sweetened beverages daily compared to those with lower consumption (three or fewer servings per month) [38]. Combined with the findings from Table 3, a higher folate intake appeared to offer protective effects against all-cause mortality in individuals with moderate to high sugar consumption. The absence of statistical significance in those with low sugar intake may be attributed to the limited sample size in this subgroup."

We believe these revisions address your suggestion and have provided a more comprehensive evaluation of the impact of dietary folate and sugar intake on mortality risk. We appreciate your insightful comments, which have helped enhance the rigor of our analysis.

Reviewer #2:

1. A paragraph summarizing the clinical take-home message of this manuscript should be added to the discussion.

Reply：Thank you for your thorough review and valuable suggestions. We have added a paragraph to the discussion section that summarizes the clinical points of this article: 

“This research underscores the potential advantages of higher dietary folate intake in reducing overall mortality among American adults with NAFLD. Healthcare providers might consider folate as a modifiable dietary factor that could enhance long-term outcomes for this patient population. Although a significant reduction in cardiovascular disease mortality was not observed, the general trend toward lower all-cause mortality indicates that dietary recommendations promoting increased folate consumption may offer a beneficial strategy for improving prognosis in NAFLD patients. Incorporating adequate folate intake into dietary guidelines and management plans could help mitigate the associated risks of NAFLD. However, additional studies are necessary to clarify the mechanisms by which folate influences mortality and to determine the ideal intake levels for different patient subgroups.”

2. The references prior to 2010 could be updated with those after 2010 since they provide more up-to-date findings.

Reply：Thank you for your suggestion regarding the references. We have carefully reviewed and updated the references cited in the manuscript. Specifically, we replaced the references published prior to 2010 with more recent studies from 2010 onwards to ensure that our citations reflect the latest findings and advancements in the field. We believe this update enhances the relevance and timeliness of the literature supporting our study.

---

## [Decision Letter · Decision Letter 1]

6 Nov 2024

Dietary folate intake and all-cause mortality and cardiovascular mortality in American adults with non-alcoholic fatty liver disease: Data from NHANES 2003 to 2018

PONE-D-24-27398R1

Dear Dr. Bai,

We’re pleased to inform you that your manuscript has been judged scientifically suitable for publication and will be formally accepted for publication once it meets all outstanding technical requirements.

Kind regards,

Aleksandra Klisic

Academic Editor

PLOS ONE

Additional Editor Comments (optional):

Reviewers' comments:

Reviewer's Responses to Questions

**Comments to the Author**

1. If the authors have adequately addressed your comments raised in a previous round of review and you feel that this manuscript is now acceptable for publication, you may indicate that here to bypass the “Comments to the Author” section, enter your conflict of interest statement in the “Confidential to Editor” section, and submit your "Accept" recommendation.

Reviewer #1: All comments have been addressed

Reviewer #2: All comments have been addressed

2. Is the manuscript technically sound, and do the data support the conclusions?

Reviewer #1: Yes

Reviewer #2: (No Response)

3. Has the statistical analysis been performed appropriately and rigorously? 

Reviewer #1: Yes

Reviewer #2: (No Response)

4. Have the authors made all data underlying the findings in their manuscript fully available?

Reviewer #1: Yes

Reviewer #2: (No Response)

5. Is the manuscript presented in an intelligible fashion and written in standard English?

Reviewer #1: Yes

Reviewer #2: (No Response)

6. Review Comments to the Author

Reviewer #1: Thank you for the review of the revised version.

All comments and suggestions have been adequately addressed.

Reviewer #2: (No Response)

7. PLOS authors have the option to publish the peer review history of their article (what does this mean?). If published, this will include your full peer review and any attached files.

Reviewer #1: **Yes: **Giuseppe Potrick Stefani

Reviewer #2: No

---

## [Editor Report · Acceptance letter]

12 Nov 2024

PONE-D-24-27398R1 

PLOS ONE

Dear Dr. Bai, 

I'm pleased to inform you that your manuscript has been deemed suitable for publication in PLOS ONE. Congratulations! Your manuscript is now being handed over to our production team.

Kind regards, 

on behalf of

Dr. Aleksandra Klisic 

Academic Editor

PLOS ONE